# Fusing Priori and Posteriori Metrics for Automatic Dataset Annotation of Planar Grasping

**Hao Sha    Qianen Lai    Hongxiang Yu    Rong Xiong    Yue Wang**[*]
Control Science and Engineering of Zhejiang University
`{shahao, laiqe, hongxiangyu, rxiong, ywang24}@zju.edu.cn`
[*] is the corresponding author

**Abstract:** Grasp detection based on deep learning has been a research hot spot in recent years. The performance of grasping detection models relies on high-quality, large-scale grasp datasets. Taking comprehensive consideration of quality, extendability, and annotation cost, metric-based simulation methodology is the most promising way to generate grasp annotation. As experts in grasping, human intuitively tends to make grasp decision based both on priori and posteriori knowledge. Inspired by that, a combination of priori and posteriori grasp metrics is intuitively helpful to improve annotation quality. In this paper, we build a hybrid metric group involving both priori and posteriori metrics and propose a grasp evaluator to merge those metrics to approximate human grasp decision capability. Centered on the evaluator, we have constructed an automatic grasp annotation framework, through which a large-scale, high-quality, low annotation cost planar grasp dataset GMD is automatically generated.

**Keywords:** Grasp Detection, Grasp Metric, Grasp Dataset, Automatic Annotation

## 1 Introduction

Grasping serves as a basic yet important skill in the robotic manipulation domain since its reliability is always the prerequisite for the following complex tasks. To equip robots with the ability to implement feasible grasps in a scene, grasp detection becomes a critical problem. For 4 Degree of freedom (DoF) planar grasp detection, learning-based algorithms have been the mainstream, which leads to intense demands for large-scale, high-quality, low annotation cost grasp datasets. There are mainly three kinds of methodologies for grasp dataset generation: hand-annotated methodology, physical trial methodology, and metrics-based simulation methodology. The hand-annotated method means that annotations are given directly by experts or crowd-sourcing according to their experience, such as the typical Cornell dataset [1]. Grounded on the hypothesis that human intuitively knows what a successful grasp look like, the hand-annotated dataset is believed with high quality. However, the huge consumption of human labor and time limits the dataset scale. Physical trial methodology means that annotations are collected from grasp attempt results through semi-automatic trials on real robots [2, 3]. Although close to ground truth since the training set and testing set share the same domain, real robotic grasping trial: i) relies on solid device support (such as the robot farm [3]) or huge time consumption; ii) is always bounded with specific robot setup leading to difficulties in generalization; iii) cannot be fully automatized which limits its scale, since human interventions are needed for specific cases. Metrics-based simulation methodology means calculating various grasp metrics [4] in simulation to evaluate the quality of candidate grasp as annotations for synthetic data. Even though the reality gap degrades its accuracy and generalization performance, low cost and high efficiency in extension make the metrics-based method popular [5, 6, 7]. Taking comprehensive consideration of quality, extendability and annotation cost, the metrics-based simulation method is the most promising way to generate grasp annotation [8] (Figure 1 shows the comparison).

To improve the quality of the dataset generated through metric-based simulation methodology, more efforts should be put into grasp metrics to mimic human grasp decisions. Human tends to make grasp decision based both on the priori knowledge (physical analysis) and the posteriori knowledge (previous experience). Similarly, grasp metrics are reorganized and split into priori and posteriori

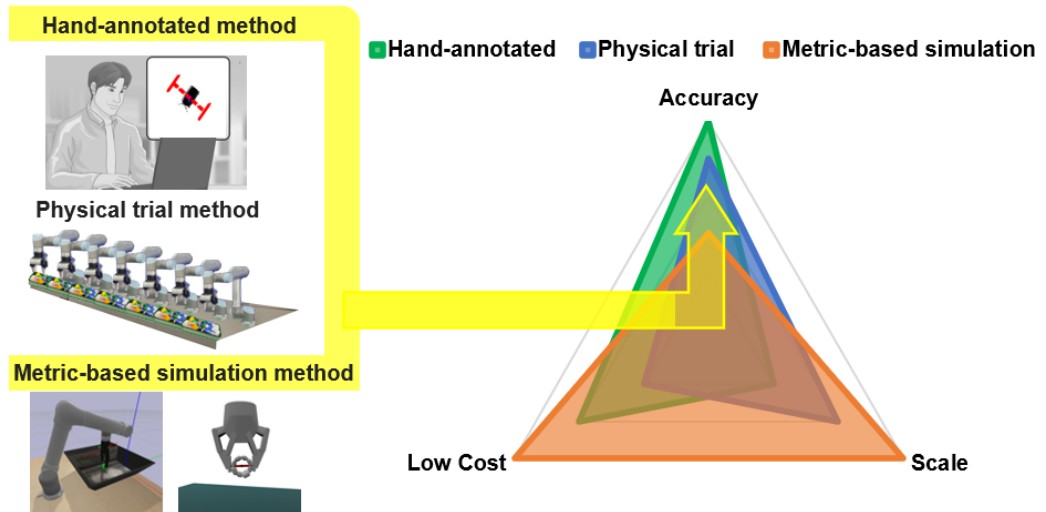

Figure 1: Qualitative comparison between three kinds of data annotation methodologies. The yellow arrow signifies that our newly proposed method aims at improving the accuracy of the existing annotation methods among metric-based methodologies.

metrics in this paper. Priori metrics are defined as metrics measured before execution based on physical analysis of object geometry, and quasi-static properties, such as the $\epsilon$-metric [9]. Although efficient in computing and strongly supported by physical theories, priori metrics only reflect partial information and ignore dynamic effects involved in real grasping. Posteriori metrics are defined as metrics concluded from grasp trial performances in simulation, implicitly offering far ampler information about the dynamic performance of grasping than priori metrics. Comparison between priori and posteriori metrics in [10] shows the latter is closer to human judgment since its consideration of reality. However, drawbacks of posteriori metric are as followed: i) High computation cost; ii) Risk of introducing noise and spurious correlations (a specific clarification is in Appendix A.1). Previous planar grasping annotating method in simulation adopt either pure prior analytical metrics [11, 5] or pure posterior performance-based metrics [12, 6] to annotate datasets. Since both individual priori and posteriori metric have their drawbacks, the sticking point is how to integrate them to get a trade-off between ampleness and comprehensiveness of information, spurious correlations, and calculation cost when using the metrics-based method.

In this paper, we build up a hybrid grasp metric group and propose a metric-based grasp evaluator to approximate human grasp decision capability. Centered on the evaluator, we have constructed an automatic grasp annotation framework, through which a large-scale, high-quality, low annotation cost planar grasp dataset GMD is generated. Through such a process, the human grasping experience can be firstly recorded and mapped into grasp metric space and then re-conveyed into the GMD dataset, which ensures that data annotations mimic human decisions. Our GMD dataset consists of 28k objects, 140k depth images, and 1.12million grasp annotations. Due to the extendability of our automatic annotation framework, the scale can be conveniently extended with new objects source. Two levels of experiments demonstrate that benchmark models trained on GMD have a higher success rate in both simulation and reality than the contrastive datasets while generating well in different domains.

## 2    Related Work

**Pure priori metrics-based datasets generation.** [13] use the form closure metric to annotate the dataset through Graspit! [14] simulator. Dex-Net2.0 dataset [5] is synthesized according to robust $\epsilon$-metric, which calculate the expectation of traditional $\epsilon$-metric [9] under preset sensor and control noises. Grasp-Net [15] adopts the improved force-closure metric to annotate 1 billion grasp samples on point clouds collected from real-world scenes, which take the minimum friction coefficient needed for keeping grasp in force closure as the quality criterion. Single metric only reflects par-

tial aspects of grasp, researchers begin to find a function combining the multi kinds of single priori metric. Qin et al. [16] define the antipodal score (similar to force closure metric), occupancy score (the volume of an object within the gripper closing region), collision score (probability of collision with other objects) and try to integrate them by the minimum operation as the final metric. Zhang [17] imposes antipodal score and center score as the metrics to generate the Regard dataset. The former refers to [16], and the latter describes the distance from the grasp to the geometric center of the object.

**Pure posteriori metrics-based datasets generation.** [18] examine that posteriori metrics-based labels were easier for DNN to learn from since single priori metric is noisier. Thus, posteriori metrics become popular for synthetic dataset generation. Johns et al. [12] used DART physical simulator to generate a grasping dataset, but it's not released publicly. Similarly, Depierre et al. [6] take grasp attempts result in Pybullet simulator as the posteriori metric to generate the Jacquard dataset with 50k pictures and about 1.1M grasp annotations. Eppner et al. [19] mainly take metrics of grasp trial result and the robustness to external shaking disturbance into consideration to automatically generate a large-scale dataset for 6-dof grasp detection.

**Combination.** The previous grasp annotation method either adopts pure prior analytical metrics or posterior metrics, how to get an optimal combination of multi metrics is still an open problem. Rubert et al. have several precious attempts [20] [10] on merging multi priori metrics to find a global index and analyze the correlation between metrics to eliminate similar ones. Their experiments show that proper combination can achieve far higher accuracy than individual priori metrics. However, they haven't tried to involve posterior metrics in their metrics combination. This inspires us to integrate both priori and posteriori metrics for grasp dataset generation.

## 3 Metrics and Evaluator

This section first proposes a hybrid metric group containing both priori and posteriori grasp metrics to extract features of grasp candidates in Section 3.1. And then a two-step evaluator is proposed to map the metrics space to the grasp feasibility in Section 3.2 (see Figure 2 for a schematic). The evaluator is trained on a small-scale expert set and then used to generate a large-scale dataset automatically in Section 4.

### 3.1 Hybrid Metric Group

Priori metrics are the metrics measured before execution based on analysis of object geometry and mechanics of priori domain knowledge. Posteriori metrics are the metrics measured during or after grasp execution, relying on posteriori knowledge concluded from trials. Priori metrics explicitly offer object geometry, and quasi-static properties in an analytical way, while posteriori metrics implicitly offer dynamical information in an empirical way. To combine their advantages, we collect 36 metrics for grasp evaluation, including 16 priori metrics, 17 posteriori metrics, and 3 contrastive metrics evaluating the difference between priori and posteriori metrics. They cover as wide aspects as possible to approximate human grasp consideration, involving geometric relations, contact force limitations, grasp stability, robustness to disturbance, grasp comfort, etc [4]. Metrics definition, selection, and correlation analysis are shown in Appendix A.2-A.4. These metrics will be used as features for the machine learning models of the subsequent two-stage evaluator.

### 3.2 Human-Mimicking Automatic Grasp Evaluator

**First-stage: Coarse Screening Model.** Due to the huge imbalance between positive and negative samples in grasping, positive ones are far more sparse. It's expensive and unnecessary to calculate all 36 metrics for those meaningless negative samples. Therefore, a decision tree is used as a coarse screening filter to accelerate the process of metrics calculation. Since decision results can be given without obtaining all features in advance, lots of meaningless negative samples can be quickly eliminated through only few metrics, obviously improving calculation efficiency. To accomplish this task better, we further improve the original decision tree in two aspects: i) introduce metric sampling cost punishment into typical information gain ratio during model training to sort metrics through a trade-off between classification performance and sampling cost (details in the equation (3) in the Appendix B.2); ii) conduct the maximum recall pruning during model testing to avoid false-negative

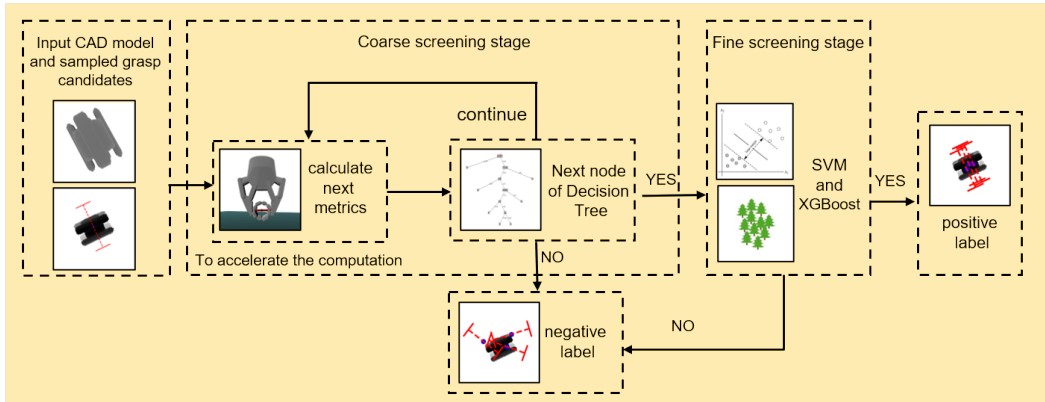

Figure 2: The proposed grasp evaluator consists of coarse and fine screening models. For each grasp candidate, we sample each grasp metric according to the nodes order of the decision tree, until arriving at leaf nodes. The positive grasp determined by the coarse screening model will be sent to the fine screening model to make a more precise judgment as the final label.

cases which are irremediable in subsequent steps. Details about model improvement and training are shown in Appendix B.2.

**Second-stage: Fine Screening Model.** After the coarse screening process, those ambiguous samples are left, which need a more accurate model to do the fine screening. Thus, we adopt a joint model composed of XGBoost [21] and SVM [22]. The inputs of the models are the feature vectors of 36 grasp metrics while the outputs are the probabilities that the grasp is feasible with preset thresholds to define the final labels. To utilize the complete 36 metrics as features for training, we conduct supplementary sampling to fill up the metrics ignored by the decision tree of those coarsely screened positive samples from the first stage. In the inference process, the prediction result depends on the voting between the two models. Those Grasps accepted by both models are labeled as the final positive samples. Details about model training are shown in Appendix B.2. According to performance on the training set, we set relatively high thresholds for the models to ensure the precision and conservativeness of the grasp decision.

**Expert Set Collection.** To train the proposed two-stage grasp evaluator, we collect a small-scale hand-annotated expert set as the ground truth. Learning from an expert set ensures the high quality of the dataset generated by the evaluator. The final expert set consists of 256 depth images and 2400 grasp annotations. Details are shown in Appendix B.1.

## 4 GMD Dataset

This section introduces the pipeline of our automatic grasp annotation framework centered on the proposed grasp evaluator and metric group (see Figure 3 for a schematic).

### 4.1 Definition

**Grasp representation.** GMD dataset offers planar grasp for parallel-jaw grasp from the top-down view, thus all of the grasps in this paper are parallel to the desktop. We define the grasp as:

$$g_{3d} = (p, \phi, w) \in \mathbb{R}^3 \times S^1 \times \mathbb{R}^1 \qquad (1)$$

where $p = (x, y, z) \in R^3$ is the position of the grasping center point in Cartesian space, $\phi \in S^1$ is the angle between the grasping axis and X-axis of the world coordinate system, $w$ is the width when the parallel-jaw is closed on objects. Then grasp annotation conducted in image space can be acquired by the following transformation:

$$g_{anno} = T_{IC} \left( T_{CW} \left( g_{3d} \right) \right) = (p, \phi, w, q) \qquad (2)$$

where $T_{CW}$ transforms from the world frame to the camera frame, $T_{IC}$ transforms from the 3D camera frame to 2D image coordinates, and $q$ is the corresponding binary label of grasp feasibility.

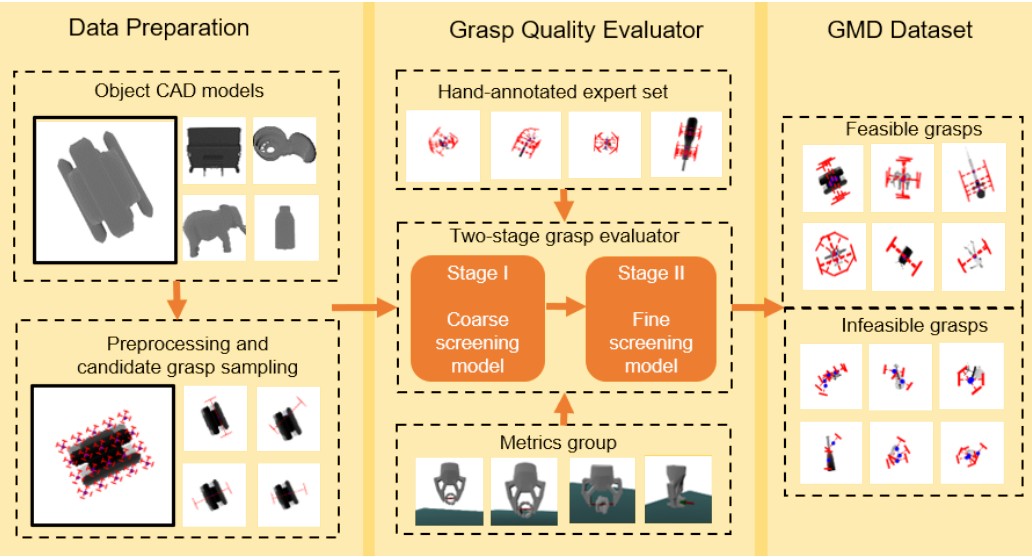

Figure 3: The key components of our automatic data generation framework are shown. The source CAD models are firstly loaded in the simulation environment and densely sampled for grasp candidates. Then, the two-stage evaluator based on the hybrid metric group automatically annotates the grasp candidates to generate the GMD dataset.

**Candidate Grasp Sampling Space.** As for each candidate grasp $g_{samp}$ in the sampling space $G_{samp}$, $z_{samp}$ is set as a constant far smaller than the height of the object, which means grasping at the near bottom of the objects. Therefore, we only need to consider the projection of both the grasp axis and the object convex hull onto the $xy$ plane, namely $Line_g^{xy} \cap Hull_O^{xy} \neq \emptyset$. Since the convex hull of the objects is inside the bounding box, the range of $G$ can be amplified to $Line_g^{xy} \cap BBox_O^{xy} \neq \emptyset$. In this way, the candidate grasp sampling space $G_{samp}$ can be obtained. For all $g_{samp}$ in the $G_{samp}$: $(x_g, y_g) \in B, z_g = D, \phi_g \in [-\pi/2, \pi/2)$, where the grasping center sampling space $B$ is obtained by extending the length and width of $BBox_O^{xy}$ by the maximum opening width of gripper.

## 4.2 Data Preparation

**Objects Preparation and Simulation Scene Generation.** We use 20k object CAD models as the object sources to generate the GMD dataset. For each object, we conduct the normalization processing. Details are shown in Appendix C.1. The Pybullet physics simulator [23] is used to build up a grasping platform for the subsequent grasping simulation and datasets synthesis.

**Candidate Grasp Dense Sampling.** To cover the candidate grasping set $G$ as much as possible, a dense sampling method is used. In the grasping sampling space $g_{samp}$ defined in Section 4.1, the center point $p$ is uniformly sampled with a step length of 5mm, the grasping depth $D$ is given to be 0.1m, and the opening width of the gripper $w$ is temporary set to be 0.1m, which will finally be determined by subsequent grasping attempts. Meanwhile, grasp angles $\phi$ are split into 20 portions and uniformly sampled for each grasp center point, and a candidate grasp set $G^*$ is obtained. On average, 7000 candidate grasps will be obtained on each object.

## 4.3 Data Annotation and Analysis

Grasp candidates are annotated by the two-stage evaluator with binary feasibility labels. The final positive samples are offered totally by the fine screening model, while the final negative samples are offered half-and-half from two stages after the scale alignment with positive samples. We used the trimesh [24] and pyrender [25] tools to render 5 depth image ($600 * 600$) for each grasp scene from multiple random perspectives with an inclination angle limited in $10°$. Then, we use the camera matrix to project all grasps from sampling space $g_{3d}$ into the image space (defined in Section 4.1). In total, the GMD dataset contains 28k objects, 140k depth images, and 1.12million grasp annotations. Table 1 shows that we have the largest-scale source objects among typical datasets for planar grasp,

which offer great diversity in object geometry. Although the total grasp scale is not the largest because of the annotation sparsity, we don't regard it as a flaw. Instead, annotation sparsity mimics the rigorousness and conservativeness of human grasp behavior on unknown objects, which may improve model performance across different domains (analyzed in Section 5.3). A small part of GMD dataset visualization can be seen in Figure 3, more of them are shown in Appendix F with Figure 8. The analysis of annotation cost can be seen in Appendix C.2.

Table 1: Summary of the properties of the famous public planar grasp datasets.

| Dataset | Total objects | Total images | Total grasps | Modality | Annotation methodology |
|---------|---------------|--------------|--------------|----------|------------------------|
| Cornell | 240 | 1035 | 8019 | RGB-D | Hand-annotated |
| Dex-Net2.0 | 1500 | 6.7M | 6.7M | Depth | Metric-based (priori) |
| Jacquard | 11K | 54K | 1.1M | RGB-D | Metric-based (posteriori) |
| GMD (ours) | 28K | 140K | 1.12M | Depth | Metric-based (priori&posteriori) |

## 5 Experimental Results

In this section, we conduct experiments on two levels. The experiments of the first level are aimed at testing the evaluator proposed in Section 3.2 to show that our evaluator considering both priori and posterior metrics can approximate human grasp decisions better than pure prior or posterior evaluator. The experiments of the second level are aimed at evaluating the GMD dataset generated by the proposed automatic annotation framework. We choose Cornell, Dex-Net2.0, and Jacquard as contrastive datasets and benchmark them on two representative grasp detection models. The benchmark model choices are: for grasping scoring method, GQCNN [5] is chosen; for one-stage grasp detection method, GR-ConvNet [26] is chosen. The model training details are described in Appendix D. Three kinds of experiments are conducted to examine the performance of the trained models: i) cross-dataset evaluation; ii) simulation experiment evaluation; iii) physical experiment evaluation (shown in Figure 4).

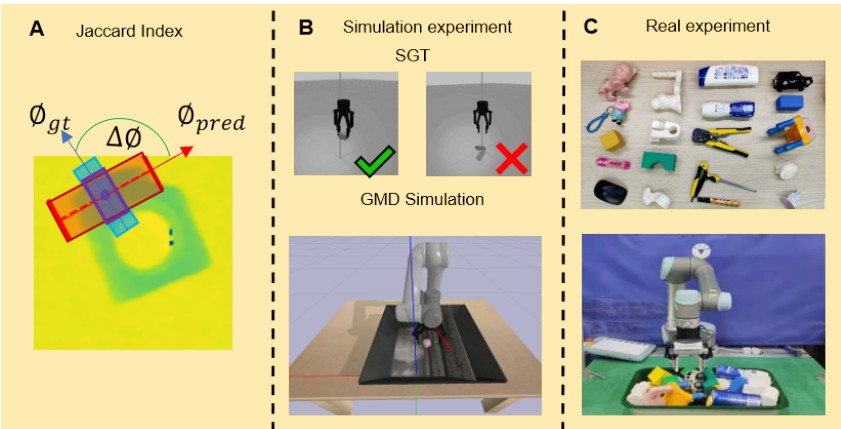

Figure 4: Three kinds of experimental setups in the second level experiment. A. The Jaccard index used for Cross-dataset Evaluation; B. SGT [6] and our GMDSIM simulative grasp criterion for Simulation Experiments; C. The real test object set and experiment platform of Real Robot Experiments.

### 5.1 First Level Experiment: Evaluator Comparison

Although many previous works take grasp metrics as ground truth to annotate datasets, few of them have compared the grasp decision generated by metrics with ground truth labeled by humans. Thus, we choose the same posteriori metric (success rate) with Jacquard [6], and the same priori metric ($\epsilon$-quality) with Dex-Net2.0 [5] to build two evaluators based on a single metric respectively. We split the expert set from Section 3.2 into a training set and a testing set. For every single metric-based grasp evaluator, we first calculate the best threshold according to ROC on the training set and

then make them a classifier. Contrastive experiments on the testing set are conducted to compare the performance of our coarse screening model, fine screening model, and two single metric evaluators. Results shown in Table 2 prove that our coarse and fine models based on hybrid metrics can better approximate human decisions than pure priori and posteriori metrics used to generate Dex-Net2.0 and Jacquard datasets.

Table 2: Comparison between different metric-based evaluators.

| Evaluator | Coarse model | Fine model | $\epsilon$-quality | sim-success |
|---|---|---|---|---|
| Accuracy (%) | 94.1 | 99.5 | 69.3 | 67.6 |

## 5.2 Second Level Experiment

**Cross-dataset Evaluation.** To evaluate whether the benchmark grasp detection model trained on GMD can generate well on other datasets, we use the typical rectangle metric [1] to do the cross-dataset evaluation. The Jaccard Index considers a grasp as correct if: i) its rotation error with ground-truth is less than 30° and ii) the rectangle IOU with ground-truth is larger than 0.25 (Figure. 4. A). For the testing set, 100 objects not involved in the training set are chosen from each dataset. The average results are shown in Table 3, which demonstrate that: i) models trained on GMD can generalize well to other datasets; ii) models trained on other datasets can only perform well on themselves and suffer from a sharp performance drop on other datasets.

Table 3: Cross-dataset evaluation based on Jaccard Index.

| GQCNN | | | | | GR-ConvNet | | | |
|---|---|---|---|---|---|---|---|---|
| test\train | Cornell | Jacquard | Dex-Net2.0 | GMD (ours) | test\train | Cornell | Jacquard | GMD (ours) |
| Cornell | 63.0 | 45.6 | 62.3 | 72.6 | Cornell | 66.3 | 57.2 | 63.0 |
| Jacquard | 81.8 | 74.8 | 87.5 | 86.4 | Jacquard | 65.5 | 69.6 | 61.3 |
| GMD (ours) | 51.5 | 35.4 | 51.9 | 78.4 | GMD (ours) | 63.1 | 56.0 | 74.2 |

**Simulation Experiments.** For the simulation experiment, the SGT criterion is used, which is adopted in Jacquard [6]. We build up a similar testing platform called GMDSIM in the Pybullet simulator (shown in Figure. 4. B), utilizing a different scene set from the data generation scene in Section 5. As for the testing set, 170 unseen object models are selected from the Shapenet dataset. Each object is grasped for five attempts in simulation to calculate the success rate. To examine the generalization to different simulation domains, both SGT with 100 testing objects and GMDSIM test are conducted. The average success rates are shown in Figure 5. A, which demonstrates that: i) models trained on GMD achieve high performance in GMDSIM test and relatively high performance in SGT test, which means that it can adapt well to other domains. ii)models trained on Jacquard can only grasp well in SGT and suffers from a sharp performance drop in GMDSIM, which imply that datasets generated by pure posteriori metric may cause over-fitting; Meanwhile, models trained on Dex-Net2.0 have higher success rates than Jacquard but lower than GMD, which imply that pure priori metric can help overcome over-fitting but the combination of priori and posteriori is the best choice.

**Real Robot Experiments.** For the real robot experiment, the experiment platform is built up as shown in Figure 4.C. The testing set is composed of regular, adversarial and household objects, totally 25 objects (more details about the setup are in Appendix E.1). Each object is grasped with 5 attempts. For each attempt, object position and pose will be reset with random in-plane rotation and shift. An attempt will be regarded as a successful grasp when the object can be lifted and transferred to a specific area without dropping. The real-world experiment results in Fig 5. B shows that: i) models trained on GMD achieve higher performance on all three kinds of objects; ii) models trained on Jacquard have low performance on three kinds of real objects, which may be caused by the reality gap introduced by the use of pure posteriori metric. iii) models trained on Cornell only have a good performance on regular objects, but are not good at adversarial objects with complex geometry, which may owe to the lack of diversity limited by scale; iv) models trained on Dex-Net2.0 are not good at household and adversarial objects, which may also suffer from reality gap caused by use of pure priori metric but better than the pure posteriori.

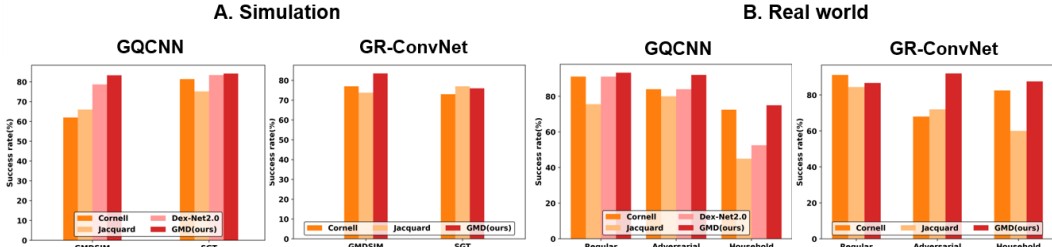

Figure 5: Simulation and real robot experiments results between Cornell, Jacquard, Dex-Net2.0, and GMD benching on GQCNN and GR-ConvNet.

**Disscusion.** To summarize, models trained on GMD are proven with high success rates in grasping unknown objects and robustness to noises, which is credited to the diversity of objects and comprehensive consideration of both priori and posteriori metrics. It also reveals that synthetic datasets generated in simulation can be used to train models for real-world grasp detection. The ability to overcome the reality gap may originate in mimicking human decision features of grasp metrics. There is an obvious phenomenon that models trained on GMD always tend to choose the most conservative grasp like a human, which is always feasible and robust in both simulation and reality. For instance (shown in Figure 6), models trained on GMD tend to grasp an object in the middle part which is the most conservative behavior like a human, and the consistency can be kept regardless of rotation, shift, and noisy disturbance. In contrast, models trained on Jacquard and DexNet2.0 probably grasp at two ends of objects. Although feasible, they are more risky choices and weak at disturbance rejection. This may be helpful to generalize to different domains, which is also the advantage of GMD over Jacquard and Dex-Net2.0. On the other hand, the diversity supported by the scale is the advantage of GMD over Cornell.

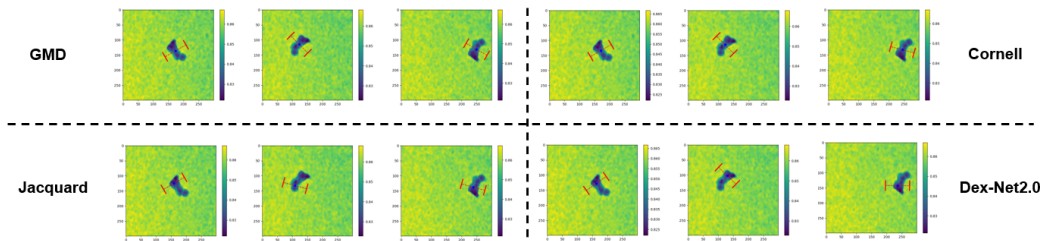

Figure 6: Decision results visualization comparison between GQCNN trained on different datasets.

## 6 Conclusion

In this paper, we build up a hybrid grasp metric group involving both priori and posteriori metrics and propose a grasp evaluator to merge those metrics to approximate human grasp decision capability. Centered on the evaluator, we have constructed an automatic grasp annotation framework, through which a large-scale, high-quality, low annotation cost planar grasp dataset GMD is generated. Two levels of experiments show that GMD annotated based on hybrid metrics is better than the dataset annotated by pure priori (Dex-Net2.0) and pure posteriori metrics (Jacquard), which is credited to the complementation between different kinds of metrics and the human mimicking properties.

Although GMD acts as a competitive dataset for planar grasp detection, some limitations remain: i) it only considers single object scenes with a simple background rather than clustering scenes with a noisy background; ii) it is only suited for top-down grasp rather than 6-dof grasp. As a general metric-based methodology for automatic grasp annotation, our hybrid metric group, grasp evaluator, and even the total annotation framework introduced in Figure 3 can naturally be migrated to generate a clustering scene and 6-Dof grasping dataset, which will be conducted in future research.

**Acknowledgments**

This work is supported and funded by the National Nature Science Foundation of China under Grant 62173293, Zhejiang Provincial Natural Science Foundation of China (LD22E050007), and Science and Technology on Space Intelligent Control Laboratory (2021-JCJQ-LB-010-13).

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
