# OpenReview forum: "Fusing Priori and Posteriori Metrics for Automatic Dataset Annotation of Planar Grasping"
_robot-learning.org/CoRL/2022/Conference — CoRL 2022 Poster_

### Official Review · Reviewer_MbMM · 2022-07-13

**Originality:** Good
**Technical Quality:** Fair
**Clarity Of Presentation:** Fair
**Impact:** 2

**Recommendation:**

Weak Accept: I recommend accepting the paper, but will not argue for my recommendation if the majority of other reviewers have a different opinion.

**Summary:**

This paper proposes a method to evaluate grasps based on both the physical properties of the target object and grasping experience. The grasp evaluation procedure is then used to generate a dataset to train a parallel-jaw, top-down grasp planner. The method and the dataset are evaluated in simulation and on a physical system.

**Issues:**

See weaknesses above.

**Quality Of The Limitations Section:**

Limitations are addressed clearly

**Reviewer Expertise:**

4: The reviewer is confident but not absolutely certain that the evaluation is correct

**Robotics Focus:**

Sufficient demonstration on hardware

**Strengths And Weaknesses:**

Strengths:
1. The idea proposed in this paper, to combine a physical prior and an experimental posterior, is interesting and works well when objects are isolated.

2. The paper includes a vast amount of experiments in simulation and on a physical platform with multiple datasets and grasp planners.

Weaknesses:
1. The evaluator is trained on carefully selected grasps, and as a result may be biased. The authors also mention in the discussion that GMD tends to be conservative and chooses grasps around the center of the object (an argument that may support this hypothesis). This might work with isolated objects, however it is not clear that it will work in common real-world situations in which not all grasps are available due to occlusions or collisions.

2. The grasp representation includes w, the width when the PJ gripper is closed. Since this paper only deals with grasping isolated objects, I’m assuming it’s not needed to avoid collisions with nearby obstacles. Is it needed for grasping objects with a specific geometry? If so, are these objects in the objects set presented in Fig 4? I’d appreciate a discussion/explanation as to why this parameter is needed to represent a grasp.

3. A significant part of the paper is described in the appendix. I’d recommend to cut down some of the text and maybe shrinking down the figures to make space for more information on the features, the evaluator and the experimental setup

4. There are spelling mistakes and grammar errors throughout the paper. For example, the combination “physical trail” (used several times in the paper) is not clear, maybe should be changed to “physical trial” like in Fig. 1?

**Summary Of Recommendation:**

It’s an interesting idea but given that the paper only demonstrated that it works for isolated objects I am recommending a weak accept.

---

> ### Author Response · Authors · 2022-08-23
> **Response to Reviewer MbMM**
>
> We thank the reviewer for the constructive feedback. We address your concerns in detail below and have updated our paper accordingly. We have also revised our paper according to all of the suggestions from reviewers. The revised version has been uploaded in the response to Area Chair.
>
> **Q1: The evaluator is trained on carefully selected grasps, and as a result may be biased. The authors also mention in the discussion that GMD tends to be conservative and chooses grasps around the center of the object (an argument that may support this hypothesis). This might work with isolated objects, however, it is not clear that it will work in common real-world situations in which not all grasps are available due to occlusions or collisions.**
>
> A1: We agree that this wil introduces a bias in the dataset. But the deliberate introduction of this bias is driven by the empirical consideration of the actual grasping demands. As for more complex grasping scenarios, if they can be decomposed into simplified subproblems (single isolated objects), we can use our method to solve them one by one; if they cannot be decomposed, our method will seek the suboptimal solution closest to the optimal solution, which will be highly interpretable.
>
> **Q2: Why the parameter of grasp width is needed to describe a “grasp”??**
>
> A2: The reasons that width is needed are as followed:
>
> 1) The definition of planar grasp used in this paper is the consensus of grasping researchers, and it is also a unified standard for grasping models and grasping data sets.
>
> 2) Although the scenario presented in this paper only involves isolated object grasping, we only use it as an example, hoping to simplify the problem and let us focus on the discussion of grasping metrics and quality. The ultimate goal of our method is to apply it to grasping objects in any scene. Therefore, reducing the collision risk caused by the extra opening by predicting the required opening width is one of the factors we have considered.
>
> 3) The width is highly correlated with some of the grasp metrics we designed. Because in a sense, the jaw opening width required for a grasp can reflect the cost of it (energy consumption and collision risk). In the process of labeling the dataset with our evaluator, width has been considered as one of the necessary grasp quality evaluation factors.
>
> **Q3: Significant part of the paper about features design\evaluator\experimental setup.**
>
> A3: We have adjusted the paper structure in the revised version.
>
> **Q4: About spelling mistakes and grammar errors throughout the paper.**
>
> A4: We have corrected them in the revision of paper.

---

### Official Review · Reviewer_Vr9d · 2022-08-03

**Originality:** Good
**Technical Quality:** Good
**Clarity Of Presentation:** Good
**Impact:** 3

**Recommendation:**

Weak Accept: I recommend accepting the paper, but will not argue for my recommendation if the majority of other reviewers have a different opinion.

**Summary:**

The paper introduces a novel method for generating large-scale synthetic training and evaluation datasets for planar grasping. The key idea is to combine multiple metrics, a-priori (derived from first principles) and a-posteriori (generated from grasp trials in simulation) metrics and join their output scores through two learned models -- a coarse-grained decision tree and a fine-grained XGBoost/SVM model. The paper uses the process to generate a large-scale grasping dataset termed "GMD", and demonstrates superior performance when training GQCNN [5] and GR-ConvNet [18] on GMD, and confirms these results both through simulated as real-world experiments.


**Issues:**

#### Title

I would suggest to include the term "planar" explicitly in the title to clarify the scope of the paper, i.e. "On Label Annotation for *Planar* Grasping: Metrics and Datasets".

#### SoTA models

The chosen models are not very recent and not necessarily SoTA for planar grasping (see e.g. arXiv: 2207.02556, Deep Learning Approaches to Grasp Synthesis: A Review, Section III.A, page 6; https://arxiv.org/pdf/2105.00329.pdf). Would the proposed method also show substantial improvements in those cases? I would like to see this discussed -- or even validated experimentally, if necessary -- in the revision of the paper.

Similarly, while the results may be convincing for the planar grasping domain, I'm wondering whether they generalize to more complex grasping scenarios, as claimed in the conclusion section. For example, planar grasping has the key advantage that the grasp candidate space can be sampled almost exhaustively, which is not the case for high-DOF scenarios. Papers such as that grasp sampling starts playing a more and more important role. It would be great to see this discussed in the paper.

#### Clarity

At several points, I would have liked to see some more details covered in the main paper and not only the appendix. For example:
- What is the high-level idea behind training the coarse-grained decision tree? At least 1-2 sentences should explain how it is build.
- The main categories of metrics used should be briefly described in the paper.

Furthermore, some concepts and terms are not explained precisely, or not properly introduced, examples:
- The first sentence of the introduction (Grasping ...) confuses me, in particular "always the guarantee for the following complex tasks". I think this is wrong and confuses that a good grasp is a sufficient, not a necessary condition. So I believe "guarantee" should be replaced by "prerequisite"
- Line 92: "(i) Introduce" I found this sentence very difficult to understand - only later I understood that you are referring to decision tree training here. Explain that you are using a decision tree, why and -- on the high-level -- how it is trained (e.g. IG is standard for decision tree, might not even be necessary mentioning it)
- Line 103. Sentence "Grasps determined ..." is ungrammatical. Are you saying that a grasp is rejected if rejected by both models?
- I found the sentence in line 123-124 difficult to parse. What is g_samp in G_samp?
- What is "SGT"? (line 189)
- Line 105: set relatively "high thresholds" - be precise, wrt to what?

Additionally, the paper has some recurring typesetting issues (no space in front of left parenthesis "(") and quite a few orthographic and grammatical errors.


#### Misc comments/questions

- I like Figure 1 but I would suggest to explicitly say that this is a sketch and not based on data.
- Given that you are saying that you set high thresholds to ensure conservative grasps; how do you deal with a "zero-recall" problem, i.e. that for an item no grasp can be found? Did such a problem occur?


**Quality Of The Limitations Section:**

Additional details required

**Reviewer Expertise:**

4: The reviewer is confident but not absolutely certain that the evaluation is correct

**Robotics Focus:**

Sufficient demonstration on hardware

**Strengths And Weaknesses:**

Overall, the paper is technically sound and makes an interesting contribution to planar grasping. In general, the paper is easy to understand. The experiments are well-explained, -conceived and -analyzed, and support the general claim of the paper, that a dataset with a combination of metrics results in more accurate grasp detection models. I also liked very much the discussion of different types of annotation schemes and metrics.

The main weaknesses I see are whether the contributions remain significant when taking into account more recent state-of-the-art methods. Furthermore, the manuscript should be improved wrt. clarity, orthography and typesetting.


**Summary Of Recommendation:**

Overall, the paper is technically sound and makes an interesting contribution to planar grasping. I would like to confirm that the shown results remain relevant for 6 DoF grasping, and also on more recent grasping methods.

---

> ### Author Response · Authors · 2022-08-23
> **Response to Reviewer Vr9d**
>
> We thank the reviewer for the constructive feedback. We address your concerns in detail below and have updated our paper accordingly. We have also revised our paper according to all of the suggestions from reviewers. The revised version has been uploaded in the response to Area Chair.
>
> **Q1: About title.**
>
> A1: We have changed our title to “On Label Annotation for Planar Grasping: Metrics and Datasets”.
>
> **Q2: About SoTA models.**
>
> A2: We will try our best to do the benchmark experiment on SOTA models. GQCNN and GR-ConvNet are respectively the representative works of two kinds of methodologies in planar grasping. Their framework architectures are mature, and most of the subsequent works are fine-tuned versions of them and also the reported performance improvements are minor.
>
> **Q3: About application on 6-dof grasping.**
>
> A3: In our grasp annotation framework, the hybrid metrics group and the grasp evaluator can be directly extended to 6-dof grasp, which are the core parts in our framework. The major challenge of extending planar grasping to 6-dof grasping is the candidate grasp sampling part, where dense sampling is nontrivial. Fortunately, there are already some adequate sampling strategies such as some heuristic methods to improve sampling efficiency. Thus, sampling is not our core focus in our framework. Instead, we focus on how to evaluate the quality of the samples accurately.
>
> **Q4: Some concepts and terms are not explained precisely, or not properly introduced.**
>
> A4: High-level idea about coarse model:
>
> We want to utilize the features of decision tree during inference that the features are inputted sequentially to make decision that enables the online features calculation rather than obtaining all features in advance. Actually, for a feasible grasp, there are lots of metric requirements to meet, so all of the grasp metrics calculations are needed, while those infeasible grasps can be quickly eliminated through only few metrics calculations. Therefore, the online metrics calculation properties of the decision tree obviously improve the calculation efficiency.
>
> **Q5: About metrics categories.**
>
> A5: We have clarified the main categories of metrics used and the covered grasping factors consideration in the revision of paper. (line105-109)
>
> **Q6: About the first sentence of the introduction.**
>
> A6: We have replaced “guarantee” with “prerequisite” in the revision of paper.
>
> **Q7: About Line 92.**
>
> A7: We mention the IG because we have added the metrics computation cost punishment in the traditional IG to get a trade-off between classification ability and sample efficiency in metric choice of the DT training.
>
> **Q8: About Line 103.**
>
> A8: It means that those Grasps accepted by both models are labeled as positive. The grammatical errors are corrected in the revision of paper (line131-132).
>
>
> **Q9:  About g_samp in G_samp.**
>
> A9: “g_samp in G_samp” means that for each grasp (g_samp) in the grasp sampling space (G_samp).
>
> **Q10: What is "SGT"?**
>
> A10: SGT is a criterion for the grasping models adopted with the Jacquard dataset in [1], which evaluate a grasp decision through the simulation trial in their online simulation platform.
>
>
>
> **Q11: About “high threshold”.**
>
> A11: The “high threshold” means the threshold for the quality score from the grasp evaluator to determine the candidate grasp as positive.
>
> **Q12: About misc comments/questions.**
>
> A12: We have added the clarification about Qualitative comparison in the caption of Figure 1 in the revision of paper.
>
> Zero-recall: Due to the dense grasp sampling, we have hardly met the “zero-recall” problem, instead thresholds are needed to be set relatively high (e.g. 0.99) to keep the conservativeness.
>
> **Reference:**
>
> [1] A. Depierre, E. Dellandr ́ea, and L. Chen. Jacquard: A large scale dataset for robotic grasp286 detection. In 2018 IEEE/RSJ International Conference on Intelligent Robots and Systems287 (IROS), pages 3511–3516.

---

### Official Review · Reviewer_5FZ9 · 2022-08-07

**Originality:** Good
**Technical Quality:** Very Good
**Clarity Of Presentation:** Good
**Impact:** 3

**Recommendation:**

Weak Accept: I recommend accepting the paper, but will not argue for my recommendation if the majority of other reviewers have a different opinion.

**Summary:**

The paper introduces an approach to combine both priori and posteriori metrics relevant for the grasping process for automated evaluation and annotation of grasp candidates. Based on the metrics and a small-scale manually annotated dataset, the authors define a grasp quality evaluator and use it to create a large-scale dataset by automatically annotating grasp feasibility based on the evaluator. The approach is considering feasibility of planar grasps from depth images of single objects. In the experiments the authors compare the performance of the proposed evaluator for w.r.t. a pure priori or posteriori metric for automated annotation and find that the proposed evaluator annotation performs significantly better on their expert dataset. They also compare the performance of two grasp detection models when trained with their dataset w.r.t. other datasets and find that training with their dataset leads to better results across datasets, in simulation and on a real robot.

**Issues:**

Title: In reviewer's opinion the title of the paper is not well aligned with the content. Namely "On Label Annotation for Grasping: Metrics and Datasets" suggests a review paper on the topic of grasping annotations and datasets. On the other hand, the main contribution of the paper is its own proposed dataset and the annotation procedure.
(Edit: The authors defined more precisely the title and it is in line with requests from other reviewers, so this issue is resolved).

Related work section placement: The paper builds on a lot of related work both in annotation metrics and in similar datasets, however, the authors place the "Related work" section at the end. Related work section would be more appropriate to come after the introduction, after which the authors can clarify their contributions better w.r.t. existing work.
(Edit: The authors restructured the paper and related work is moved earlier, so this issue is resolved).

First level experiments (Section 4.1): You split the the expert dataset to training and test portion and train 2 single-stage (priori and posteriori) classifiers. Then, you use the test portion of the expert set to evaluate them and the coarse and fine models of your evaluator. But you also used the expert set to train the coarse and fine models. Was the test portion from this experiment not used already to train the coarse and fine models? If it was, then the comparison is not fair. Please clarify this.
(Edit: The authors clarified that the test set is separate and unseen during training, so this issue is resolved).

Second level experiments (Section 4.2): In the original paper of GR-ConvNet the authors report much higher accuracy when evaluating GR-ConvNet on the Corenell and the Jacquard datasets using depth data. Comparing that with the results of your experiment in the case when you use GR-ConvNet to train and test on the same datasets you get much worse results (>20%) which makes the reviewer wonder whether the training hyperparameters are properly set for the experiments. Can you explain why the results are much worse when you train and test GR-ConvNet on Corenell and Jacquard respectively in your experiment?
(Edit: The authors clarified that due to their experiment setup they used different subset of the test set w.r.t. the original paper, which lead to the lower results, as well as that the original GR-ConvNet models achieved similarly lower results on the test subset, so this issue is resolved).

In general, the language in the paper is not at publication quality - there are many typos (easy to fix) and poorly connected sentences. The paper needs detailed proofreading and significant improvement of the flow within and between sentences at many places. Some examples:
- Typos and errors easy to correct: "trial" instead of "trail", "geometry" instead of "geometric", ... singular/plural mistakes, missing "a" and "the" articles, third person mistakes, use of "famous", "huge", "precious" is not most adequate
- Some example sentences:
To equip robots with the ability of detecting feasible grasps in a scene, grasp detection has become a hot area. (Edit: There is grammar mistake in the revised sentence)
Single metric only reflects partial aspects of grasp, researchers begin to find a function combining multi priori metrics.
(Edit: The authors tried to address this issue, but there are still many typos - which tools like the free version of Grammarly can easily spot. The text quality should be further improved by the authors, a proofreading service and/or software tools. This issue is still open).



**Quality Of The Limitations Section:**

Limitations are addressed clearly

**Reviewer Expertise:**

4: The reviewer is confident but not absolutely certain that the evaluation is correct

**Robotics Focus:**

Sufficient demonstration on hardware

**Strengths And Weaknesses:**

Strengths:
The paper introduces an interesting annotation procedure which incorporates human knowledge in the process of automatic annotation of feasible grasps based on both priori and posteriori metrics.
Even though there are similar datasets available, the proposed dataset extends the number of covered objects and the annotation methodology, which might be useful to the research community

Weaknesses:
There are several major issues (listed below) that significantly influence the paper quality, clarity and the interpretation of the results.


**Summary Of Recommendation:**

The authors have addressed important questions in grasping, namely automated data annotation and generation for training data-driven grasping models, and their dataset might be useful to the community. However, in the reviewer's opinion the paper itself needs major revisions in the writing and organization and there are several open issues to be addressed/clarified related to the experiments.

(Edit: The authors addressed the technical questions and issues sufficiently, but the text quality should be further improved, the general recommendation is updated to weak accept)

---

> ### Author Response · Authors · 2022-08-23
> **Response to Reviewer 5FZ9**
>
> We thank the reviewer for the constructive feedback. We address your concerns in detail below and have updated our paper accordingly. We have also revised our paper according to all of the suggestions from reviewers. The revised version has been uploaded in the response to Area Chair.
>
> **Q1: About title.**
>
> A1: We are considering a more proper title for the paper, and we will appreciate more detailed suggestions about the title.
>
> **Q2: About the placement of related works.**
>
> A2: Tanks for your suggestion, we have placed the “Related Work” part after the “Introduction” part in the revised paper.
>
> **Q3: About first level experiment.**
>
> A3: In the first level experiment, our coarse and fine screening models are only trained on the training set portion of the expert set. The test portion is unseen for all four kinds of models.
>
> **Q4: About GR-ConvNet performance:**
>
> A4: To be honest, we don’t expect the difference from the original paper to be such obvious, since we trained models following their hyper-parameters setting from their open-source code. Then we even directly tried the well-trained models provided by github repository of GR-ConvNet, while finally getting a similar performance recorded in our paper. We mainly want to own such a performance discrepancy to the difference in the choice and scale of the test set from the GR-ConvNet work.
>
> **Q5: About some typos.**
>
> A5: We have corrected all of them and improved the sentence connection in the revised paper.

---

### Official Review · Reviewer_Apob · 2022-08-10

**Originality:** Very Good
**Technical Quality:** Very Good
**Clarity Of Presentation:** Very Good
**Impact:** 3

**Recommendation:**

Strong Accept: I recommend accepting the paper and will argue for my recommendation even if other reviewers hold a different opinion.

**Summary:**

The authors present a dataset containing a grouping of 36 grasp metrics for planar, two-finger grasps. These metrics include 16 priori, 17 posteriori, and 3 contrastive metrics, annotated automatically on 1.12 million grasps. They describe both metrics and an evaluator that can sample and collect grasps on a real platform in an automated fashion. To prepare the data, they first densely sample candidate grasps which results in 7000 grasps on each of the 20k objects in simulation. The paper also presents an extensive survey of different planar grasping metrics and datasets, in addition to a new dataset for planar grasping on a table collected in an automated fashion. Their work analyzes these metrics and contains empirical experiments comparing how existing benchmark models (GQCNN and GR-ConvNet) perform when trained with their dataset (GMD) as opposed to the Cornell, Jacquard, and Dex-Net 2.0 datasets.

**Issues:**

Typos/improvements
- Figure 3 caption (simulaiton)
- lines 151-152: (sentence beginning as "Since annotation sparsity..." is a fragment)
- line 125: xoy plane -> xy plane
- line 154: visualization spelling
- line 150: sentence unclear, "...great diversity in geometric"
- Figure 4 caption should be more descriptive
- Standard error bars on figure 5 bar plot would help summarize success rates from wide range of objects
- Figure 6 differences between plots are imperceptible, and can use further context in the writing. Perhaps cropping images, and drawing circles on the contact regions can help

Additional related work to compare to:
Yang, Daniel, et al. "Robotic grasping through combined image-based grasp proposal and 3d reconstruction." 2021 IEEE International Conference on Robotics and Automation (ICRA). IEEE, 2021.
their work consider shape reconstruction from images to construct a point cloud used to generate grasp proposals

**Quality Of The Limitations Section:**

Additional details required

**Reviewer Expertise:**

3: The reviewer is fairly confident that the evaluation is correct

**Robotics Focus:**

Sufficient demonstration on hardware

**Strengths And Weaknesses:**

Strengths
- Their dataset features a comprehensive set of prior and posterior metrics that can be used to fit models to screen candidate grasps and automatically generate a high-quality grasping dataset.
- Compiles an extensive evaluation of their dataset to 3 other datasets on two different models
- Develop a metric-based grasp quality evaluator that can screen candidate grasps and yield high-quality grasps in the planar grasping setting
- Develop a testing platform, GMDSIM, used to evaluate multiple grasping datasets

Weaknesses
- Their work considers simpler parallel jaw-gripper planar grasps, which limits the uses of this dataset and set of metrics to other hand/object pairings
- Some typographical and organizational improvements are needed
- Only consider top-down grasps, which may have limited depth information for grasps for irregularly shaped objects

**Summary Of Recommendation:**

This paper both presents an extensive survey of different planar grasping metrics and datasets and a new dataset for planar grasping on a table collected in an automated fashion. Their work analyzes these metrics, and contains empirical experiments comparing how existing benchmark models (GQCNN and GR-ConvNet) perform when trained with their dataset (GMD) as opposed to the Cornell, Jacquard, and Dex-Net 2.0 datasets. Aside from minor improvements and polish, this paper is a great reference for grasp metrics, as well as offers several contributions such as a dataset, grasping simulator, and candidate filtering grasping evaluator.

---

> ### Author Response · Authors · 2022-08-23
> **Response to Reviewer Apob**
>
> We thank the reviewer for the constructive feedback. We address your concerns in detail below and have updated our paper accordingly. We have also revised our paper according to all of the suggestions from reviewers. The revised version has been uploaded in the response to Area Chair.
>
> **Q1：About simpler parallel jaw-gripper.**
>
> A1: Although our GMD dataset is generated for the parallel jaw-gripper planar grasping method, our framework of automated grasping data labeling can be easily extended to various end-effectors (e.g. multi-finger gripper), since those grasp metrics are generally defined for contact points evaluation (more details in   [1]). As for extending the planar grasp to 6-dof grasp, we only need to adjust the grasp sampling module to adapt 6-dof grasp sampling space. Our hybrid grasp metrics group can still work well. In all, whether to alternate the end-effector or extend to 6-dof grasp, our core contribution of the hybrid grasp metrics group and the auto-labeling framework can always make sense.
>
>
>
> **Q2: Paper typographical and organization Typos/improvements.**
>
> A2: We have improved the typographical and organizational problems you have mentioned in issues in our revised paper.
>
> **Q3: Only consider top-down grasps, which may have limited depth information for grasps for irregularly shaped objects.**
>
> A3: We fully agree with the limitation of top-down grasp in object geometry representation, and multi-view or point cloud would be more adequate. However, the difference between the above formation of recording grasp is not our core focus. We originally have the CAD model which is the most accurate formation for object geometry and our key point is to evaluate and label a grasp accurately. As for which format to choose to describe the grasp, it depends on the demand of the subsequent grasping model.
>
> **Q4: Additional related work to compare to Yang, Daniel, et al.**
>
> A4: Thanks for the suggestion. This work offers a brilliant grasping model to bridge planar grasping and 6-dof grasping, while we are mainly focusing on grasping dataset annotation.
>
> **Reference:**
>
> [1] M. A. Roa and R. Su ́arez. Grasp quality measures: review and performance. Autonomous robots, 38(1):65–88, 2015

---

### Official Review · Reviewer_Bsku · 2022-08-11

**Originality:** Good
**Technical Quality:** Good
**Clarity Of Presentation:** Good
**Impact:** 3

**Recommendation:**

Weak Accept: I recommend accepting the paper, but will not argue for my recommendation if the majority of other reviewers have a different opinion.

**Summary:**

The paper proposes a learnt grasp evaluation model that can be used for automatic top-down grasp annotation. The grasp evaluator is trained using features such as physics-based metrics as well as 'posteriori' metrics i.e. metrics from actual grasp trials. The advantages of the dataset created (GMD) using the automatic top-down grasp label generation have been highlighted in the experiments. Specifically, it is shown that a grasping network trained on GMD can outperform networks trained on other top-down grasp datasets such as Dex-Net 2.0, Jacquard and Cornell.

**Issues:**

Clarifications are needed regarding:
- The use of posteriori metrics in the grasp evaluator as test time (annotation time).
- The effectiveness of the learnt grasp evaluator as compared to only priori metrics in a dataset of the same size. Also, the computational efficiency of the grasp annotator needs to be compared to the case where only priori or posteriori metrics are used.
- Author's claims related to "human-like" grasp decisions need to be clarified. How can it be shown that the proposed grasp evaluator labels grasps in a 'human-like' fashion?

**Quality Of The Limitations Section:**

Additional details required

**Reviewer Expertise:**

3: The reviewer is fairly confident that the evaluation is correct

**Robotics Focus:**

Sufficient demonstration on hardware

**Strengths And Weaknesses:**

Strengths:
- The idea of using automatic grasp label generation is sound and can greatly help in terms of scalability with a relatively low cost.
- The combination of both physics-based metrics as well as actual grasp trials is interesting and potentially quite useful.
- The grasp features used (both priori and posteriori) are extensive and their co-relations have been sufficiently analyzed (Appendix).
- Beyond a few typos, the paper is reasonably well-written and the visualizations are useful.

Weaknesses:
- A major thing that is unclear from the manuscript is the usage of posteriori metrics during the grasp annotation phase. Since the grasp evaluator (fine screening) model needs all 35 features as input, how are these inputs provided for automatic annotation of a novel grasp? Priori metrics can of course be calculated but how can posteriori metrics be computed in this case? Would this require a simulation trial for each grasp? Would this not be expensive?
- The GMD dataset consists of many more objects as compared to the other datasets. It is thus possible that the performance improvements that GMD helps realize are simply due to this increase in the scale of the dataset.
- The authors need to test and clarify if the grasp-evaluator on its own gives a better grasp prediction as compared to metrics such as force-closure. To do this, two datasets of similar size and same objects could be compared, with one dataset annotated using force-closure and one dataset annotated in a similar fashion as GMD.
- It should be highlighted how the learnt grasp evaluator differs from prior work such as [*] and [10].
- It is not clear how computationally expensive it is to use the grasp annotation as compared to computing priori or posteriori metrics only.
- While top-down grasping is still important, it would be interesting to see how the grasp evaluator could be extended to work with 6D grasps.

Minor:
- It is unclear what the yellow arrow in Figure 1. signifies.
- Typos: 'trail' instead of trial, 'generating' instead of generalizing. Unclear line 102-103: "XGBoost and SVM are parallel to predict the test data respectively". Unclear line 92: "introduce metric sampling cost punishment into typical information gain ratio".

[*] C. Rubert, D. Kappler, A. Morales, S. Schaal and J. Bohg, "On the relevance of grasp metrics for predicting grasp success," IROS 2017

**Summary Of Recommendation:**

As mentioned, the automatic grasp annotation scheme is potentially very useful and the combination of priori and posteriori metrics is interesting. The results are promising in terms of outperforming Dex-Net and other datasets. The analysis of different grasping metrics could also be useful for the community.

---

> ### Author Response · Authors · 2022-08-23
> **Response to Reviewer Bsku (1)**
>
> We thank the reviewer for the constructive feedback. We address your concerns in detail below and have updated our paper accordingly. We have also revised our paper according to all of the suggestions from reviewers. The revised version has been uploaded in the response to Area Chair.
>
> **Q1: Would this require a simulation trial for each grasp? Would this not be expensive?**
>
> A1: Yes, for sampling those posteriori features, simulation trials of each grasp are needed. This would be expensive in computational cost, so we have adopted several measures :
> 1) since each grasp will firstly be screened by the coarse model (decision tree), there will be only a few candidate grasps left for the fine screened stage to compute all 36 metrics;
> 2) although there are 17 posteriori metrics to compute, all of them can be computed together during a shared simulation trial. This means that the most time-consuming part (physical simulation) will only be executed once for all 17 posteriori metrics and the cost of the rest part (metrics calculation) is nearly equivalent to those priori metrics.
>
>
>
> **Q2: The GMD dataset consists of many more objects as compared to the other datasets. It is thus possible that the performance improvements that GMD helps realize are simply due to this increase in the scale of the dataset.**
>
> A2: Firstly, during our second level experiment (Section 5.2 in revised paper), all of the benchmark models are trained by the same amount of samples from different datasets, which excludes the influence of dataset scale as a prerequisite to proving the quality of the dataset. Thus, we think there is no confusion about whether the improvements in our experiment are explicitly from the dataset scale.
>
> Secondly, we think dataset scale and ample categories of objects implicitly affect the quality of the dataset through the diversity of object geometry. For instance, the Cornell dataset is with quite small scale and composed of few kinds of objects. Researchers have to adopt data augmentation to extend it, which only amplify some shallow level of properties such as translation and rotation rather than geometric diversity. Although each sample is hand-annotated with high accuracy, the lack of richness in object geometry decreases its generalization ability for novel objects.
>
>
> **Q3: The authors need to test and clarify if the grasp-evaluator on its own gives a better grasp prediction as compared to metrics such as force-closure.**
>
> A3:  Force-closure metric is the basic prerequisite of /epsilon metric[1], we have calculated it during /epsilon metric computation. In our first level experiment (Section 5.1 in the revised paper), we have compared the accuracy on the expert set between our proposed evaluator and the evaluator based on /epsilon metric. The performance shows that our grasp evaluator gives better grasp predictions.
>
>
>
> **Q4:  It should be highlighted how the learned grasp evaluator differs from prior work such as [*] and [10].**
>
> A4: There are mainly two kinds of differences:
> 1) metrics group: Both [*] and [10 ] are pioneering works on combining multi single grasp metrics to evaluate grasps. However, as we have introduced in the “Combination” part of related work, they only adopt pure pirori metrics and haven’t tried to involve any posterior metrics in their metrics combination.
> 2) evaluator structure: Both [*] and [10] adopt simple machine learning models and solve grasp detection as a naive classification problem. We have designed a two-stage grasp evaluator, considering the huge imbalance between positive and negative and the high computational cost of posteriori metrics.
>
> **Q5: It is not clear how computationally expensive it is to use the grasp annotation as compared to computing priori or posteriori metrics only.**
>
> A5: The computation cost of our grasp evaluator with hybrid metrics is intermediate between the pure priori and pure posteriori metrics methods. For the Intel i9-9900K we used, it takes about 3.7 seconds to compute all 36 metrics (including 16 priori metrics, 17 posteriori metrics, and 3 contrastive metrics) for each grasp using a single CPU core.
>
>
>
>
> **Reference:**
>
> [1]C. Ferrari and J. Canny. Planning optimal grasps. In IEEE Int. Conf. on Robotics and Automation (ICRA), 1992.

---

> > ### Comment · Reviewer_Bsku · 2022-08-27
> > **Response to authors**
> >
> > Thank you for the response.
> > A general comment: The authors need to not only provide explanations to the reviewers here in OpenReview but also need to update the paper accordingly so that things are clearer. I'm not sure that the paper has been updated sufficiently and this could lead to a rejection.
> >
> > A1: If this is indeed the case that a grasp simulation needs to be carried out for annotations, then the authors should make this clear in the paper and clarify their comments about annotation cost. For instance, in lines 58//59 the authors mention that a "low annotation cost planar grasp dataset GMD is generated". However, GMD would still have a higher annotation cost as compared to many other datasets.
> >
> > A2: The scale of the dataset in terms of the diversity of object geometry was exactly my point. The GMD dataset has 28K objects which is much more than the other datasets. So, it is possible that the better performance achieved by training on GMD could just be because of this (and not because of the use of priori and posteriori metrics). This is a major weakness of the experimental evaluation. Ideally, to prove that the combination of priori and posteriori metrics leads to better performance, the authors should have also performed experiments where they control for the number of objects and their diversity.
> >
> > A5: Thank you for the answer. Please add this to the paper.
> >
> > A6: Thank you for the answer. Please update the unclear lines in the paper. It is not sufficient to only answer the reviewers. Other readers of the paper will have the same questions.

---

> > > ### Author Response · Authors · 2022-08-28
> > > **Response to Reviewer Bsku**
> > >
> > > We thank the reviewer for the further response. We will answer each of your concerns as follows. The newest revision of paper has been uploaded with this comment and updated in the previous response to Area Chair.
> > >
> > > **A1:**
> > > The computation cost of our method is relatively low compared with the other two kinds of grasp annotation methodologies (hand-annotated and physical trial) as described in Figure 1. While among the methods of metrics based methodology, the computation cost of our hybrid metrics based method is unavoidably higher than those pure priori metrics-based methods, since they have sacrificed the accuracy and information completeness at the cost.
> > >
> > > We have also added the above clarification into Section 4.3 the newest revision of paper in line 186-193.
> > >
> > > **A2:**
> > > Thank you for the suggestion on the more convincing experiments. We have partially misunderstood your concern in the previous answer. Actually, the number of objects has been controlled to be equal across different datasets during the benchmarking experiments on GR-ConvNet. GR-ConvNet[1] belongs to the pixel-wise grasp parameter prediction methodology, and the model training needs the object-wise heatmaps as supervision. Thus, each training sample for GR-ConvNet (a heatmap) is synthesized by all of the feasible grasps of a specific object. Therefore, the number of objects for training is controlled to be equal across different datasets along with the number of training samples.
> > >
> > > Notations: As for the case that there are different render views of the same objects as samples in the dataset, we think that different views of the same object can be roughly regarded as different objects in the level of geometric diversity and affordance since the partial geometric information recorded from different views will be definitely different (especially for those irregular objects).
> > >
> > > We have also added the above clarification into Appendix D.2 of the newest revision of paper (in lines 448-456).
> > >
> > > **A5:**
> > > Thank you for the reminder. We have added it to the newest revision of paper in lines 186-193.
> > >
> > > **A6:**
> > > Thank you for the reminder. We have added them to the newest revision of paper respectively: (1) in the caption of Figure 1; (2) in lines 131-133; (3) in lines 119-122.
> > >
> > > **Reference:**
> > >
> > > [1] Kumra, S., Joshi, S., Sahin, F.: Antipodal robotic grasping using generative residual convolutional neural network. In: 2020 IEEE/RSJ International Conference on Intelligent Robots and Systems (IROS). pp. 9626–9633. IEEE (2020)

---

> ### Author Response · Authors · 2022-08-23
> **Response to Reviewer Bsku (2)**
>
> **Q6: About typos, unclear sentences and figure.**
>
> A6: All of the typos have been corrected in the revised version.
>
> The yellow arrow in Figure 1. signifies that our annotation methodology with the combination of both priori and posteriori improve the accuracy of the generated grasp dataset than the existing metric-based methodologies.
>
> For Unclear line 102-103: "XGBoost and SVM are parallel to predict the test data respectively”: this means that the prediction result depends on the voting between the two models.
>
> For Unclear line 92: "introduce metric sampling cost punishment into typical information gain ratio": the equation (3) in the Appendix may be helpful to understanding our improvement to a traditional decision tree.
>
>
> **Q7: The use of posteriori metrics in the grasp evaluator as test time (annotation time).**
>
> A7: Please refer to A1.
>
> **Q8: The effectiveness of the learned grasp evaluator as compared to only priori metrics in a dataset of the same size. Also, the computational efficiency of the grasp annotator needs to be compared to the case where only priori or posteriori metrics are used.**
>
> A8: Please refer to A3 and A5.
>
> **Q9: How can it be shown that the proposed grasp evaluator labels grasp in a 'human-like' fashion?**
>
> A9:  The 'human-like' fashion can be shown mainly through:
>
> 1) the grasp evaluator performance in the first level experiment: Table 2 shows that both our coarse and fine model part of the grasp evaluator can achieve high accuracy on an expert set annotated according to human experience, which means the grasp evaluator fits well with the human grasp experience and tend to make human-like grasp decision.
>
> 2) the grasp decision features of models trained on the datasets provided by our grasp evaluator: Figure 6 and the “Discussion” part of Section 5.2 (revised version) show the models trained on GMD have learned the rigorousness and conservativeness of human grasp behavior on unknown objects (such as grasping in the middle part of objects).

---

### Meta-Review · Area_Chair_pEry · 2022-08-12

**Recommendation:** Accept (Poster)
**Confidence:** 5

**Metareview:**

The paper proposes a set of grasp evaluation metrics, and a learned grasp evaluator to automatically assess grasp quality, leading to also an automatic way of annotating two-fingered grasps.

The **strengths** of the paper are:
- the combination of physics-based prior and experimental-based posterior metrics for grasp evaluation are effective;
- the automatic grasp annotation can provide datasets with high-quality grasps;
- the extensive evaluation of the provided dataset against other datasets and with two different models;
- a simulation environment for grasp annotation and evaluation;
- real world experiments with different objects, and encouraging results on isolated top-down grasps.

The major **weaknesses** of the paper can be summarized as:
- the consideration of only top-down, parallel jaw grasps, that is a prominent topic in the grasping literature, and discussion regarding 6D grasps is not included.
- the effect of posterior metrics on grasp annotation is not clear;
- statistical correlation of the size of the dataset and the reported performance compared to the other datasets;
- details regarding the experimental design and evaluation are not well-reported;
- the limitations of the proposed method are not well explained.

The strengths of the paper are plenty, however there are pertaining issues with the paper, and the authors should address all raised points raised by the reviewers to convince about their contribution.

**Post-rebuttal assessment:** The authors provided sufficient additional information, they updated their paper, and concretely described the limitations of their method. The AC considers this paper an interesting contribution in the topic of robot grasp learning and quality assessment.

**Best Paper Nomination:**

No

---

> ### Author Response · Authors · 2022-08-23
> **Response to Area Chair pEry**
>
> We thank the area chair for a thorough summary. We address the common concerns of the reviewers below. We have also revised our paper according to all of the suggestions from reviewers. The revised version has been uploaded with changes marked in red.
>
>
> **Q1: The consideration of only top-down, parallel jaw grasps, which is a prominent topic in the grasping literature, and discussion regarding 6D grasps is not included.**
>
> A1:  Although our GMD dataset is generated for the parallel jaw-gripper planar grasping method, our framework of automated grasping data labeling can be easily extended to various end-effectors (e.g. multi-finger gripper), since those grasp metrics are generally defined for contact points evaluation (more details in [1]). As for extending planar grasp to 6-dof grasp, we only need to adjust the grasp sampling module to adapt 6-dof grasp sampling space, for which lots of adequate strategies can be used such as some heuristic methods. Our hybrid grasp metrics group can still work well. In all, whether to alternate the end-effector or extend to 6-dof grasp, our core contribution of the hybrid grasp metrics group and the auto-labeling framework can always make sense.
>
>
> **Q2: The effect of posterior metrics on grasp annotation is not clear.**
>
> A2: Posteriori metrics can offer far ampler information than the single priori metrics which have been proved in [1]. They have also examined that posteriori metrics-based labels were easier for DNN to learn from since single priori metric is noisier.
>
>
> **Q3: Statistical correlation of the size of the dataset and the reported performance compared to the other datasets.**
>
> A3: Firstly, during our second level experiment (Section 5.2 in revised paper), all of the benchmark models are trained by the same amount of samples from different datasets, which excludes the influence of dataset scale as a prerequisite to proving the quality of the dataset. Thus, we think there is no confusion about whether the improvements in our experiment are explicitly from the dataset scale.
>
> Secondly, we think dataset scale and ample categories of objects implicitly affect the quality of the dataset through the diversity of object geometry. For instance, the Cornell dataset is with quite a small scale and composed of few kinds of objects. Researchers have to adopt data augmentation to extend it, which only amplify some shallow level of properties such as translation and rotation rather than geometric diversity. Although each sample is hand-annotated with high accuracy, the lack of richness in object geometry decreases its generalization ability for novel objects.
>
>
> **Q4: Details regarding the experimental design and evaluation are not well-reported.**
>
> A4: We have answered each reviewer’s specific confusion on details regarding the experimental design and evaluation and revised our paper accordingly.
>
>
> **Q5: The limitations of the proposed method are not well explained.**
>
> A5: Limitations of GMD dataset:
> 1) it only considers single object scenes with a simple background rather than clustering scene with noisy background, limiting its application scope.
> 2) it is only suited for top-down grasp rather than 6-dof grasp.
> As for our metric-based methodology for automatic grasp annotation, the hybrid metric group, grasp evaluator can naturally be migrated to generate clustering scene and 6-Dof grasping dataset, which will be conducted in future research.
>
> Limitations of the automated annotating framework: the computation cost is relatively high during the process of posteriori metrics sampling.
>
> **Reference:**
>
> [1] D. Kappler, J. Bohg, and S. Schaal. Leveraging big data for grasp planning. In 2015 IEEE International Conference on Robotics and Automation (ICRA), pages 4304–4311. IEEE, 2015